# Volatile Components, Antioxidant and Phytotoxic Activity of the Essential Oil of *Piper acutifolium* Ruiz & Pav. from Peru

**DOI:** 10.3390/molecules28083348

**Published:** 2023-04-10

**Authors:** Carmela Fiorella Cuadros-Siguas, Oscar Herrera-Calderon, Gaber El-Saber Batiha, Najlaa Hamed Almohmadi, Nada H. Aljarba, José Alfonso Apesteguia-Infantes, Eddie Loyola-Gonzales, Freddy Emilio Tataje-Napuri, José Francisco Kong-Chirinos, José Santiago Almeida-Galindo, Haydee Chávez, Josefa Bertha Pari-Olarte

**Affiliations:** 1Department of Pharmacology, Bromatology and Toxicology, Faculty of Pharmacy and Biochemistry, Universidad Nacional Mayor de San Marcos, Jr Puno 1002, Lima 15001, Peru; carmelacuadros9@gmail.com (C.F.C.-S.); japesteguiai@unmsm.edu.pe (J.A.A.-I.); 2Department of Pharmacology and Therapeutics, Faculty of Veterinary Medicine, Damanhour University, Damanhour 22511, Egypt; gaberbatiha@gmail.com; 3Clinical Nutrition Department, College of Applied Medical Sciences, Umm Al-Qura University, Makkah 24381, Saudi Arabia; nhmohmadi@uqu.edu; 4Department of Biology, College of Science, Princess Nourah bint Abdulrahman University, Riyadh 11671, Saudi Arabia; nhaljarba@pnu.edu.sa; 5Department of Pharmaceutical Science, Faculty of Pharmacy and Biochemistry, Universidad Nacional San Luis Gonzaga, Ica 11001, Peru; eddie.loyola@unica.edu.pe; 6Departamento de Ciencias Comunitarias, Facultad de Odontología, Universidad Nacional San Luis Gonzaga, Ica 11001, Peru; freddy.tataje@unica.edu.pe; 7Department of Surgical Clinical Sciences, Faculty of Human Medicine, Universidad Nacional San Luis Gonzaga, Ica 11001, Peru; jose.kong@unica.edu.pe; 8Department of Basic Sciences, Faculty of Human Medicine, Universidad Nacional San Luis Gonzaga, Ica 11001, Peru; santiago.almeida@unica.edu.pe; 9Department of Pharmaceutical Chemistry, Faculty of Pharmacy and Biochemistry, Universidad Nacional San Luis Gonzaga, Ica 11001, Peru; hchavez@unica.edu.pe (H.C.); bertha.pari@unica.edu.pe (J.B.P.-O.)

**Keywords:** essential oil, GC-MS, *Piper acutifolium*, herbicides, allelopathic, matico

## Abstract

*Piper acutifolium* Ruiz & Pav is known as “matico” and belongs to the Piperaceae family, and in Peru it is traditionally used as an infusion or decoction to ameliorate wound healings or ulcers. In this study, the aim was to investigate the volatile components, the antioxidant profile, and the phytotoxic activity of the essential oil (EO) of *P. acutifolium* from Peru. To identify the phytoconstituents, the EO was injected into a Gas Chromatography-Mass Spectrometry (GC-MS) to obtain the chemical profile of the volatile components, followed by the antioxidant activity carried out by the reaction with three organic radicals (2,2-diphenyl-1-picrylhydrazyl (DPPH); 2,2′-azinobis-(3-ethylbenzothiazoline)-6- sulfonic acid (ABTS); ferric reducing/antioxidant power (FRAP)). Finally, the phytotoxic capabilities of the EO were tested on two model plants, *Lactuca sativa* seeds and *Allium cepa* bulbs. As a result, the analysis identified α-phellandrene as its main volatile chemical at 38.18%, followed by β-myrcene (29.48%) and β-phellandrene (21.88%). Regarding the antioxidant profile, the half inhibitory concentration (IC_50_) in DPPH was 160.12 ± 0.30 µg/mL, for ABTS it was 138.10 ± 0.06 µg/mL and finally in FRAP it was 450.10 ± 0.05 µg/mL. The phytotoxic activity demonstrated that the EO had high activity at 5% and 10% against *L. sativa* seed germination, the inhibition of root length, and hypocotyl length. Additionally, in *A. cepa* bulbs, the inhibition root length was obtained at 10%, both comparable to glyphosate, which was used as a positive control. The molecular docking on 5-enolpyruvylshikimate-3-phosphate synthase (EPSPS) revealed that α-phellandrene had −5.8 kcal/mol, being near to glyphosate at −6.3 kcal/mol. The conclusion shows that the EO of *P. acutifolium* presented antioxidant and phytotoxic activity and might be useful as a bioherbicide in the future.

## 1. Introduction

Essential oils (EOs) are a kind of natural product obtained from the different organs of aromatic plants (seeds, leaves, fruits, stems, roots) by different methods, both conventional and unconventional, such as steam distillation, hydrodistillation, solvent extraction, supercritical fluids, and microwaves, among others. [1]. The chemical composition of EOs varies according to the plant genus, soil type, altitude, plant part extracted, growth conditions, etc. [2]. The composition of EOs generally contains compounds identified as monoterpenes and sesquiterpenes which have demonstrated several biological activities such as insecticidal, antitumor, cytotoxic, antimicrobial, anti-inflammatory, hepatoprotective, gastroprotective, anxiolytic, antidepressant, phytotoxic, and analgesic, etc. [3]. 

In recent decades, a growing number of phytosanitary pesticides have been administered to various agricultural regions. There is research on the possible use of essential oils as bio-pesticides in sustainable agronomy [4]. The Food and Drug Administration of the USA includes EOs in the category of Generally Recognized as Safe [5]. On the contrary, essential oils rich in monoterpenes have shown toxicity assessed in vitro and in vivo studies, so their use should be strictly controlled by rigorous toxicity studies. The phytotoxic effect of EOs has been associated with the presence of volatile compounds such as carvacrol, camphor, α-pinene, 1,8-cineole, limonene and thymol, which have varied individual phytotoxicity levels [6]. 

Regarding the use of EOs as a bioherbicide, several reports consider its use to treat invasive plants that contaminate crops. The herbicides can be selective (2,4-dichlorophenoxyacetic acid), non-selective (glyphosate), or work during different stages of plant development [7]. Numerous biochemical processes contribute to its herbicidal activity, including the disruption of the cuticle and damage to young tissues, the inhibition of photosynthesis and mitochondrial respiration, the change in enzymatic and phytohormone regulation, the modification of membrane properties and interactions, and the induction of microtubule disruption, genotoxicity, and reactive oxygen and nitrogen species. [8]. 

Antioxidants are molecules that can react with free radicals, reducing their oxidizing power, causing oxidative stress and slowing or retarding oxidation. These oxidizing agents are very harmful when there is an imbalance in the oxidation-reduction system and can affect lipids, proteins, and carbohydrates in organisms [9]. Regarding the antioxidants and the phytotoxic activity, an inverse relationship exists because the toxicity induced in weeds by bioherbicides can be directly associated with the generation of reactive oxygenated species to produce damage [10]. However, in recent years, the antioxidant activity of essential oils has been studied to prevent food from oxidation during storage, in food stabilizers, and in active packaging and edible coatings in the food industry [11]. Thus, the investigation of antioxidants in essential oils seems to be focused on the inhibition of lipid peroxidation, free radicals, or chelating metal ions, which are processes involved with several pathological conditions in humans. Furthermore, depending on its extraction method and other pretreatments prior to the distillation process, the antioxidant activity can provide different results [12]. 

In Peru, *Piper acutifolium* Ruiz & Pav (Piperaceae Family) is commonly known as *matico,* and a decoction of its leaves (approximately 50 g per liter of water) is used as a wash for sores and ulcers, as well as for vaginal infections [13]. It is also recommended for gastritis and menstrual disorders, with an oral administration of two cups per day [14]. According to Lognay et al., its essential oil is abundant in β-ocimene, α-copaene, allo-aromadendrene, α-cadinene, δ-cadinene, myristicin, and dillapiole [15]. A study showed that benzoic acid derivates isolated from the dichloromethane extract had activity against *Leishmania* spp., *Trypanosoma cruzi*, and *Plasmodium falciparum* [16]. There are no reported studies based on its antioxidant and its phytotoxic activity. Therefore, it is necessary to characterize the essential oil obtained by gas chromatography coupled to mass spectrometry (GC-MS) and to determine the antioxidant activity in vitro, and the phytotoxic activity in lettuce seeds and onion bulbs, which might be useful as a potential bioherbicide in the future.

## 2. Results and Discussion

### 2.1. Chemical Characterization of the Essential Oil of P. acutifolium

The obtained EO of *P. acutifolium* presented a yellow color (golden) with an extraction yield of 0.067%, a relative density of 0.9834 ± 0.03 g/mL at 20 °C, and a refractive index of 1.5000 at 20 °C. The profile chemical identified by GC-MS is reported in Table 1. 

The volatile organic compounds identified by GC-MS revealed 14 compounds (Table 1), two of which had unknown structures. The GC-MS revealed the presence of alpha-phellandrene as the major component at 38.18%, followed by β-myrcene (29.48%) and β-phellandrene (21.88%). Although there are not many reports about its composition, Lognay et al., found that *P. acutifolium* from Cajamarca, Peru was abundant in β-ocimene (8.1%), followed by β-caryophyllene (7.9%), δ-cadinene (6.8%), and α-cadinene (6.7%) [15]. Additionally, there are a variety of compounds that are dominant in different *Piper* species and the essential oils within this genus vary from monoterpene and sesquiterpenes such as in *P. demeraranum* which has β-elemene (33.1%) and limonene (19.3%); in *P. duckei* trans-caryophyllene (27.1%), α-muurolol (17.9%), and germacrene D (14.7%) [17]. In *P. aleyreanum*, *P. anonifolium*, and *P. hispidum* were β-elemene (16.3%), selin-11-en-4-α-ol (20.0%), and β-caryophyllene (10.5%), respectively [18]. In *P. aduncum*, *P. arboreum* and *P. tuberculatum* leaves from Brazil, the most abundant components were linalool (31.7%), bicyclogermacrene (49.5%), and β-caryophyllene (40.2%), respectively [19]. *P. corcovadensis* contains 30.62% 1-butyl-3,4-methylenedioxybenzene [20]. In *P. umbellatum* from Costa Rica and Brazil is germacrene D (34.2–55.8%) [21]. A study by Cicció of the *P. augustum* from Costa Rica found that α-phellandrene (14.7%) was the most abundant chemical constituent [22]. According to Ruiz-Vasquez et al., some species from the *Piper* genus from the Peruvian Amazonia was the main component in *P. coruscans* (β-bisabolene; 33.4%), *P. tuberculatum* (β-bisabolene; 40.2%), *P. casapiense* (caryophyllene oxide; 10.2%)*, P. obliquum* (bicyclogermacrene; 7.9%), *P. dumosum* (germacrene-*D; 10.4%), P. anonifolium* (caryophyllene; 11.3%), *P. reticulatum* (apiol; 15%)*, P. soledadense*, (limonene; 38.5%), *P. sancti-felicis* (apiol; 76.1%), *and P. mituense* (apiol; 51.6%) [23]. Table 1 differs from other studies and might be due to a special chemotype from the Peruvian flora. Furthermore, climate, soil, and other edaphic factors could be responsible for the chemical composition of the *P. acutifolium* from Cajamarca, Peru. 

**Table 1 molecules-28-03348-t001:** Chemical composition of the essential oil of *P. acutifolium*. % refers to the relative percentage of each compound based on a mean of three determinations. LRI ^Ref^ is the linear retention index obtained from the database [24]; LRI ^Exp^ is the linear retention index calculated against *n*-alkanes C9–C24.

#	Compound	LRI ^Ref^	LRI ^Exp^	%	MolecularFormula/Molecular Mass	ChemicalStructure	Chemical Group
1	α-Pinene	950	948	2.82	C_10_H_16_(136.23)	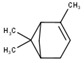	Monoterpenehydrocarbon
2	β-Myrcene	960	958	29.48	C_10_H_16_(136.23)	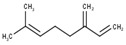	Monoterpenehydrocarbon
3	α-Phellandrene	972	969	38.18	C_10_H_16_(136.23)	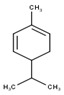	Monoterpenehydrocarbon
4	*o*-Cymene	1052	1042	1.55	C_10_H_14_(136.22)	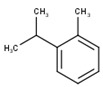	Aromaticmonoterpenehydrocarbon
5	β-Phellandrene	1082	1075	21.88	C_10_H_16_(136.23)	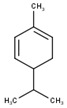	Monoterpenehydrocarbon
6	α-Gurjunene	1420	1419	0.38	C_15_H_24_(204.35)	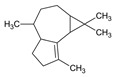	Sesquiterpenehydrocarbon
7	β-Caryophyllene	1585	1579	1.20	C_15_H_24_(204.35)	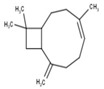	SesquiterpeneHydrocarbon
8	Humulene	1452	1579	0.26	C_15_H_24_(204.35)	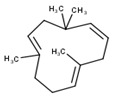	Sesquiterpenehydrocarbon
9	Germacrene D	1480	1515	0.85	C_15_H_24_(204.35)	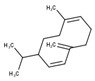	Sesquiterpenehydrocarbon
10	Bicyclogermacrene	1500	1492	0.63	C_15_H_24_(204.35)	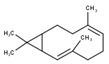	Sesquiterpenehydrocarbon
11	δ-Cadinene	1513	1510	0.94	C_15_H_24_(204.35)	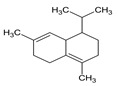	Sesquiterpenehydrocarbon
-	Unknown I	-	1600	0.78	C_15_H_26_O(222.37)	n.d.	Oxygenated Sesquiterpene
-	Unknown II	-	1623	0.27	C_15_H_26_O(222.37)	n.d.	Oxygenated Sesquiterpene
12	Aromadendrene	1656	1660	0.78	C_15_H_26_O(222.37)	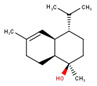	Oxygenated Sesquiterpene
Aromatic monoterpene hydrocarbons	1.55%	
Monoterpene hydrocarbons	92.36%	
Sesquiterpenes hydrocarbons	4.26%	
Oxygenated sesquiterpenes	1.83%	
Total identified	100.00%	

n.d., not determined.

### 2.2. Antioxidant Activity of the Essential Oil of P. acutifolium

According to Table 2, the results show the antioxidant activity using three organic radicals (DPPH, ABTS and FRAP). As presented, the best method to demonstrate antioxidant activity was in ABTS with an IC_50_ of 138.10 ± 0.06 µg/mL and a TEAC of 150.56 ± 8.27 µmol TE/g of EO, followed by DPPH and FRAP. Studies of antioxidant activity using the evaluated methods reveal that in non-polar plant extracts, the correlation between DPPH and ABTS is limited, and decreased in comparison to polar extracts [25]. 

Other *Piper* species showed higher levels of antioxidant activity, such as *P. auritum* (IC_50_ = 14.8 ± 0.5 µg/mL), *P. aduncum* (IC_50_ = 30.1 ± 1.8 µg/mL), and *P. umbellatum* (IC_50_ = 32.3 ± 1.3 µg/mL) against DPPH radicals [12,21,26]. Bagheri et al. [12] demonstrated that the EO obtained by hydro distillation had better antioxidant activity than supercritical CO_2_ extraction in *P. nigrum* against DPPH, with an IC_50_ of 103.28 µg/mL and 316.27 µg/mL, respectively. Alminderej et al., [27] showed that the essential oil of *P. cubeba* fruit had an antioxidant activity against DPPH and FRAP of IC_50_ = 110.00 ± 0.08 µg/mL, and 106.00 ± 0.11 µg/mL respectively. Valerazo et al., [28] reported the antioxidant activity (IC_50_) of *P. ecuadorense,* with values of more than 2.5 mg/mL for DPPH and 1.81 ± 0.09 mg/mL for the ABTS .^+^ radical. Additionally, *P. brachypetiolatum* had an IC_50_ value of 64.8 µg/mL and *P. madeiranum* had a value of 66.8 µg/mL against the DPPH radical [29]. In this study, the essential oil from *P. acutifolium* had low antioxidant activity (Table 2), which could be due to the presence of monoterpene hydrocarbons in high percentages as some essential oils with high antioxidant activity contain oxygenated monoterpenes or sesquiterpenes in their composition [30]. 

### 2.3. Phytotoxic Activity of the Essential Oil of P. acutifolium on L. sativa Seeds and A. cepa Bulbs

The use of bio-herbicides based on natural products, such as essential oils, is generating much interest in the world with regard to environmental concerns. Glyphosate is a commonly used synthetic herbicide that blocks the enzyme 5-enol-pyruvyl-shikimate-3-phosphate synthase (EPSPS) and catalyzes the sixth step in the shikimic acid pathway, reducing the aromatic amino acids such as phenylalanine, tyrosine, and tryptophan [31]. It is used as a positive control in this study, and its phytotoxic activity is presented in Table 3, Figure 1 and Figure 2, with the inhibition of seed germination, root length, hypocotyl length, and the rate of root length/stem length. In this study, the evaluation of *P. acutifolium* EO on *L. sativa* seeds (Table 3) showed phytotoxicity activity at 5% and 10%, comparable to 2% glyphosate. *P. acutifolium* EO negatively affected the germination of *L. sativa* seeds (Figure 1), depending on the concentration, while the negative control (0.1% DMSO) did not show any phytotoxic activity. However, germination was inhibited at the highest concentration, being similar to the positive control (2% glyphosate). The root and hypocotyl length of *L. sativa* was inhibited when exposed to 5% and 10% EO solutions compared with the negative control (Table 3). It is known that some volatile terpenes have revealed phytotoxic activity against *L. sativa* seeds including α-pinene, γ-terpinene, and *p*-cymene from *Eucalyptus grandis* [32]; α-pinene, and β-pinene from *Pinus brutia* [33], 1,8-cineole, β-phellandrene, and α-pinene from *Majorana* hortensis, o-cymene, and α-pinene from *Thymus vulgaris*, estragole, limonene, and β-pinene from *Carum carvi* [34], and β-caryophyllene, (Z)-caryophyllene, and germacrene D from *Ailanthus altissima* [35]. There was no significant phytotoxic effect between *P. acutifolium* EO at 5% (0.93 ± 0.12 cm; *p* = 0.9983) and 10% (0.47 ± 0.06 cm; *p* = 0.7172) with glyphosate (0.83 ± 0.15 cm) on *A. cepa* bulbs (Figure 3); the other concentrations had values of 6.33 ± 0.58 cm at 0.1% (*p* < 0.0001), 4.5 ± 0.6 cm (*p* < 0.0001) at 0.5%, and 1.93 ± 0.12 cm (*p* = 0.0151) at 1% respectively. Studies of *A. cepa* bulbs indicate that terpenes have a phytotoxic effect, such as β-pinene, δ-carene, and limonene from *Heterothalamus psiadioides,* and β-caryophyllene and spathulenol from *Baccharis patens* [35]. Abd-ElGawad et al. concluded that the phytotoxic activity of the EOs are linked to its content of oxygenated terpenoids. However, some sesquiterpenes like β-caryophyllene and derivates have demonstrated phytotoxic action, and in this study only aromadendrene at 0.78% was identified, and this oxygenated sesquiterpene might have a synergism phytotoxic activity on *L. sativa* and *A. cepa* bulbs. In addition, monoterpenes and sesquiterpenes extracted from EOs have shown phytotoxic effects causing anatomical and physiological changes in plants, some proposed mechanisms such as the accumulation of lipid globules in the cytoplasm, oxidative stress, the reduction of mitochondria, and the inhibition of DNA synthesis might be involved in its phytotoxicity. 

### 2.4. Molecular Docking Studies

Molecular docking studies were carried out in order to determine the binding affinities of α-phellandrene, β-myrcene, β-phellandrene and glyphosate with EPSP synthase protein. The binding energies (Δ*G* in kcal/mol) are listed in Table 4. The best ligand poses belonging to the population were found to be the highest in the cluster of RMSD tolerance of 0.5 and the lowest binding energy that were chosen for the docking study. A total of 95% of the ligand pose population belonging to the lowest binding energy cluster showed that the α-phellandrene occupied the deep core of the binding pocket and was involved in alkyl interactions with Ala174, Arg200, and Ile325 residues (Table 4). Rest residues were involved in non-bonded van Der Waal’s interactions, as displayed in Figure 2A. Similar interactions were also observed for beta-myrcene, where alkyl interactions were observed with Lys28, Ala174, and Arg200 (Figure 2B). β-phellandrene exhibited alkyl interactions with Ala174, Arg200, and Ile325 (Table 3, Figure 2C). On the other site, α-phellandrene exhibited the lowest binding energy Δ*G* = −5.8 kcal/mol and the highest affinity for the protein. The standard ligand glyphosate was involved in conventional hydrogen bonds with Lys28, Thr101, Ala100, Arrg405, and Arg357 residues and presented a high affinity and a binding energy Δ*G* = −6.3 kcal/mol (Table 3, Figure 2D).

### 2.5. Molecular Dynamics Simulation 

Molecular dynamics and simulation studies were carried out to determine the stability and convergence of 3FJZ-glyphosate, 3FJZ-α-phellandrene, 3FJZ-β-phellandrene, and 3FJZ-β-myrcene. The simulation of 200 ns displayed a stable conformation while comparing the root mean square deviation (RMSD) values. The RMSD of Cα-backbone of 3FJZ-glyphosate exhibited a deviation of 3.0 Å, while α-phellandrene bound protein exhibited RMSD 2.5 Å (Figure 4A). RMSD of 3FJZ-β-phellandrene exhibited 2.7 Å (Figure 4A), while 3FJZ-β-myrcene exhibited unstable RMSD up to 60 ns and later stabilized at 2.9 Å (Figure 4A). All of the RMSD values are within the acceptable range, which should be below or near 3 Å. A stable RMSD plot during simulation signifies a good convergence and stable conformations. Therefore, it can be suggested that α-phellandrene and β-phellandrene bound to 3FJZ are quite stable in complex due to the higher affinity of the ligand. The plot for root means square fluctuations (RMSF) displayed large spikes of fluctuation in 3FJZ-glyphosate are observed at 10–16, 105–110 and 152–160 residues, which may be due to the higher flexibility of the residues (Figure 4B). 1M17 with compound 2 also exhibited fluctuations at 20–54, 125–146, 150, 175–182, 230–240 and 220–230 residue positions (Figure 4B), while 3FJZ-α-phellandrene exhibited fewer significant fluctuations except at 10–30, 110–120, 150, 175–182, 230–240 and 220–230 residues (Figure 4B). The flexibility of the residues was observed to be more in the case of α-phellandrene. For β-phellandrene and β-myrcene, similar peaks were observed to that of glyphosate bound complex (Figure 4B). Most of the residues fluctuated less during the entire 200 ns simulation, indicating the rigid amino acid conformations during the simulation time. Fewer fluctuations indicate a less active protein, since more flexibility allows the protein to become functionally active. Therefore, based on the RMSF plots it can be suggested that the α-phellandrene bound protein is more flexible and becomes active during simulation in ligand-bound conformations. The radius of gyration (Rg) is the measure of the compactness of the protein. Here in this study, the 3FJZ-glyphosate Cα-backbone displayed a lowering of Rg from 19.0 to 18.5 Å (Figure 4C). On the other hand, a lowering and stable pattern was also observed for α-phellandrene, β-phellandrene and β-myrcene bound to 3FJZ from 19.0 to 18.5 and 18.6 Å, respectively, (Figure 4C). The initial lowering of Rg indicates a highly compact orientation of the protein in a ligand-bound state. A number of hydrogen bonds between the protein and the ligand suggest the significant interaction and stability of the complex. The number of hydrogen bonds between 3FJZ-glyphosate showed significant (4) numbers, and with α-phellandrene, β-phellandrene and β-myrcene, a single hydrogen bond was observed on an average (Figure 4D) throughout the simulation time of 200 ns. Followed by Rg analysis, the pattern was also observed in the solvent accessible surface area (SASA) in both the ligand-bound and unbound states. The unbound state of ligands to the 3FJZ protein (Figure 4F–H) displayed a high surface area accessible to solvents in all the cases (Figure 4E–H). The SASA value lowered as compared to the unbound state when bound with a ligand (Figure 4E–H). The study of Rg means that the binding of ligands forces the respective proteins to become more compact.

### 2.6. Molecular Mechanics Generalized Born Surface Area (MM-GBSA) Calculations

Utilizing the MD simulation trajectory, the binding free energy along with other contributing energy in the form of MM-GBSA were determined for each 3FJZ-glyphosate, 3FJZ-α-phellandrene, 3FJZ-β-phellandrene and 3FJZ-β-myrcene. The results (Table 5) suggested that the maximum contribution to Δ*G_bind_* in the stability of the simulated complexes were due to Δ*G_bind_*Coulomb, Δ*G_bind_*vdW and Δ*G_bind_*Lipo, while, Δ*G_bind_*Covalent and Δ*G_bind_*SolvGB contributed to the instability of the corresponding complexes. The 3FJZ-glyphosate, 3FJZ-α-phellandrene complexes showed comparatively higher binding free energies. These results supported the potential of glyphosate and α-phellandrene molecules with 3FJZ, and showed the efficiency in binding to the selected protein and the ability to form stable protein-ligand complexes.

### 2.7. In-Silico Toxicological Study of P. acutifolium EO

The human hepatotoxicity (HH), carcinogenicity and Ames’s toxicity analysis are presented in Table 5. Compounds 1, 4, 5, 6, 7, 8, 9, 11, 12, and glyphosate had a good safety profile. However, compounds 2, 3 and 10 might be hepatotoxic. In addition, compounds 2, 5, 8 and 9 might be carcinogenic. In a study, the oral administration of β-myrcene (#2) induced nonneoplastic lesions in the kidney of rats [36]; with regard to other compounds, there are no toxicological reports. The results for the environmental toxicity predictions of the volatile compounds of the EO fall within 1000 L/kg < BCF < 5000 L/kg categorized as bioaccumulative by the United States Environmental Protection Agency under the Toxic Substances Control Act [37]. However, glyphosate showed a low value (BCF = 0.151), which indicates that it is not bioaccumulative. Regarding the other parameters, such as IGC_50_, LC_50_FM, and LC_50_DM, the volatile components had higher values than glyphosate, indicating that these compounds isolated in the EO of *P. acutifolium* might have less toxicity when compared with glyphosate. According to the data in Table 6, some compounds can be considered toxic in silico, but the use of EO as a bioherbicide has an advantage compared to synthetic herbicides; for example, they are volatile and rapidly degrade in the ecosystem [38]. However, data on its toxicity have yet to be reported, and it will depend mainly on its chemical composition and in vivo tests.

## 3. Materials and Methods

### 3.1. Plant Material

A quantity of 9 kg of the *P. acutifolium* leaves were collected from Cajabamba, Cajamarca, Peru, located at 2658 masl (7°37′25″ S and 78°02′39″ W)) in December 2022. The material plant was authenticated by biologist Jose Campos (certificate Id. 09092022-JP). Dried and cleaned leaves (yield = 90%) were incorporated in a Clevenger apparatus to obtain the essential oil by steam distillation for 2 h. Finally, the essential oil was separated from the hydrolate by decantation, and then anhydrous Na_2_SO_4_ was added to purify the EO. Finally, the EO was kept in a sealed amber vial at 4 °C.

### 3.2. Identification of the Volatile Compounds by Gas Chromatography–Mass Spectrometry (GC–MS)

The volatile components of the essential oil of *P. acutifolium* leaves were identified using a Gas Chromatograph coupled to Mass Spectrophotometry (GCMS-QP2010 SE, Shimadzu) with a Quadrex capillary column (007 METHYL 5%) (30 m × 0.25 mm, 0.20 μm film thickness, Agilent Technologies). The EO was diluted in chloroform HPLC grade (1000 µL) (1: 100). The chromatographic conditions were set according to a previous analysis protocol [39]. The identification of volatile components was based on a comparison of relative retention indices (RIs) and mass spectra data (NIST–5 library) and published literature. Each Retention Index was calculated in comparison to a homologous series of *n*-alkanes C9–C25 (C9, 99% BHD purity and C10–C25, 99% Fluka purity). 

### 3.3. Evaluation of the Antioxidant Activity of the EO of P. acutifolium against 2,2-Diphenyl-1-Picrylhydrazyl (DPPH) Radical

The EO of *P. acutifolium* (150 µL) at different concentrations (0.1; 0.5; 1.0; 5.0 and 10.0%) reacted with 850 µL of 0.01 mM DPPH methanolic solution (Sigma Aldrich St Louis, MO, USA). The reaction lasted 30 min, and the absorbances were read at 517 nm. Each test was carried out in triplicate [40]. 

### 3.4. Evaluation of the Antioxidant Activity of the EO of P. acutifolium against 2,2′-Azinobis–(3-Ethylbenzothiazoline)-6- Sulfonic Acid (ABTS .^+^) Radical

The EO of *P. acutifolium* (150 µL) at different concentrations (0.1; 0.5; 1.0; 5.0 and 10.0%) reacted with 850 µL of ABTS radical’s solution (Sigma Aldrich St Louis, MO, USA). The reaction in dark conditions lasted 7 min, and the absorbances were read at 734 nm. Each test was carried out in triplicate [41]. 

### 3.5. Evaluation of the Antioxidant Activity of the EO of P. acutifolium against the Ferric Reducing/Antioxidant Power (FRAP)

The EO of *P. acutifolium* (150 µL) at different concentrations (0.1; 0.5; 1.0; 5.0 and 10.0%) reacted with 850 µL FRAP radical’s solution, which was elaborated with acetate buffer pH 3.6, and 2,4,6-Tris(2-pyridyl)-s-triazine diluted in HCl, and ferric chloride hexahydrate. The reaction was stopped at 4 min and the absorbances were read at 593 nm. Each test was carried out in triplicate [42]. 

### 3.6. Phytotoxicity Test on Lactuca sativa Seeds

Germination and growth bioassays were performed in Petri dishes (90 mm in diameter), with Whatman filter paper number 42. Commercial seeds of *Lactuca sativa* (lettuce) were treated with different solutions of the EO solubilized in 0.1% DMSO at 0.1; 0.5; 1.0; 5 and 10%, respectively. The Petri dishes were then incubated at 22 °C for 5 days in the absence of light to simulate the germination process. Several parameters were evaluated, including the number of germinated seeds, the root length, and the hypocotyl length. A solution of 0.1% DMSO was used as a negative control, and a 2% glyphosate solution (Fuego^®^, NeoAgro, Lima, Peru) was used as a positive control. Each Petri dish had twenty seeds, and four replicates were utilized [43].

### 3.7. Phytotoxicity Test on Allium cepa Bulbs

*A. cepa* phytotoxicity assay was carried out according to Haq et al. [44]. *A. cepa* bulbs weighing between 70–90 g were obtained from a local market for this study. Three bulbs were cut at the base to generate root growths. Finally, *A. cepa* bulbs were placed in 50 mL Falcon tubes containing different concentrations of the EO (0.1, 0.5, 1.0, 5.0 and 10.0%). The negative control was 0.1% DMSO, which was used as the diluent, and a 2% glyphosate was used as a positive control. Tubes were kept at room temperature for 5 days. After 5 days, the root lengths were measured in each evaluated concentration. The phytotoxicity assay was carried out in triplicate. 

### 3.8. Molecular Docking

This study’s purpose is to provide insight into the binding propensities of the main volatile components of *P. acutifolium* which represented the major percentages in the total composition, and these compounds were α-phellandrene, β-myrcene, and β-phellandrene, with 5-enolpyruvylshikimate-3-phosphate synthase (EPSPS) synthase (PDB ID: 3FJZ) [45]. Additionally, glyphosate was used as the standard to validate the ligand binding. Autodock vs. 4.2.6 was used to perform docking investigations on the molecules in question. The main volatile components and the protein EPSPS were saved in pdbqt format after combining non-polar hydrogens. Molecular docking was analyzed within a grid box dimension of 20.04 × 21 × 20.5 Å. The binding pocket residues were determined to form the adjacent residues at 4Å radius of the co-crystallized ligand [45]. 

### 3.9. In-Silico Toxicity of the Volatile Components of P. acutifolium

The ADMETlab 2.0 program (http://admetmesh.scbdd.com, accessed on 10 March 2023) was used to estimate the toxicity parameters and environmental toxicity. The parameters evaluated were human hepatotoxicity, the bioconcentration factor expressed as −log10[(mg/L)/(1000*MW)], the IGC_50_: concentration of the test chemical in water (mg/L) that causes 50% growth inhibition to the species *Tetrahymena pyriformis* after 48 h, the LC_50_FM that refers to the concentration of the test chemical in water (mg/L) that causes 50% of fathead minnow to die after 96 h, and the LC_50_DM: concentration of the test chemical in water (mg/L) that causes 50% of the species *Daphnia magna* to die after 48 h. These three parameters (IGC_50_, LC_50_FM, and LC_50_DM) were also expressed as −log10[(mg/L)/(1000*MW)]. To interpret the hepatotoxic, Ames’ test, and the carcinogenicity test results, they were evaluated based on probability values, where values near to 1 are highly toxic [46]. 

### 3.10. Molecular Dynamics Simulation

Molecular dynamics simulation studies of 3FJZ-glyphosate, 3FJZ-α-phellandrene, 3FJZ-β-phellandrene and 3FJZ-β-myrcene were carried out in Desmond [47] for a 200 nanoseconds time scale. Docking experiments were the first step in creating protein and ligand complexes for the molecular dynamics simulation. In static settings, molecular docking studies can predict the ligand binding state. Since docking provides a static representation of a molecule’s binding pose in a protein’s active site, MD simulations typically compute atom motions over time by integrating Newton’s classical equation of motion [48]. The ligand binding status in the physiological environment was predicted using simulations. The Protein Preparation Wizard was used to pre-process the protein–ligand complex, which included complex optimization and minimization. The System Builder tool was used to create all of the systems. As transferable intermolecular interaction potential 3 points, the solvent model with an orthorhombic box was chosen. The simulation used the OPLS 2005 force field. Counter ions were used to make the models neutral. An amount of 0.15 M salt (NaCl) was added to simulate physiological circumstances. For the entire simulation, the NPT ensemble with 300 K temperature and 1 atm pressure was used. Before the simulation, the models were relaxed [49]. The trajectories were saved for evaluation every 100 ps, and the simulation’s stability was confirmed by comparing the RMSD, Rg, hydrogen bonds, and the solvent accessible surface area of the protein and ligand over time.

### 3.11. Binding Free Energy Analysis

The molecular mechanics and generalized Born surface area method (MM-GBSA) was utilized to determine the binding free energy of standard glyphosate, α-phellandrene, β-phellandrene, and β-myrcene complexes, respectively. Using the Python script thermal mmgsba.py and the VSGB solvation model, along with the OPLS5 force field and one-step sample size, the MM-GBSA free energy of binding was measured in the final fifty frames of the simulation trajectory. The individual energy modules of columbic, covalent, hydrogen bond, van Der Waals, self-contact, lipophilic, and solvation of ligand and protein were aggregated using the concept of additivity in order to determine the binding free energy of MM-GBSA (kcal/mol). To determine the Δ*G_bind_*, the following equation was used:$$\Delta G_{bind}=\Delta G_{MM}+\Delta G_{Solv}-\Delta G_{SA}$$
where:

Δ*G_bind_* is binding free energy (kcal/mol), Δ*G_MM_* designates free energy differences of the 3FJZ + ligand complex and the total energies of 3FJZ and ligands in isolated form, Δ*G_Solv_* is the solvation energy of the ligand-receptor complex and the sum of the solvation energies of the 3FJZ and ligands in the unbound state, and Δ*G_SA_* is the surface area energy differences between 3FJZ and ligands.

### 3.12. Statistical Analysis

Graph Pad Prism v4.0 was used to introduce the results obtained in this study. An ANOVA test and Dunnett’s post-test were used as statistical tests, and a *p* value more than 0.001 was considered statistically significant in the phytotoxicity test. The Trolox equivalent antioxidant capacity (TEAC) was expressed as µmol, and the Trolox equivalent/g of EO and linear regression using Microsoft Excel was used to calculate the half inhibitory concentration (IC_50_).

## 4. Conclusions

Despite the fact that some synthetic herbicides have harmful side effects, they are becoming vital for effective weed management. Recent interest and demand for organic fruits, vegetables, dairy products, and beverages throughout the world, especially in industrialized nations, has led to an interest in bioherbicides and the search for products with lower harmful side effects. It is concluded that the EO presented 14 compounds and as the main fingerprint to β-myrcene, which was determined by GC-MS. Regarding the antioxidant activity against DPPH, ABTS and FRAP, the best result was using the ABTS radical and its antioxidant profile compared to other *Piper* species reported in the literature. Furthermore, the phytotoxic activity revealed that at 5% and 10%, the EO inhibited several parameters in the development of Lettuce seeds and in onion bulbs, such as germination and root length. The glyphosate control used in this study also showed phytotoxic activity, and its main target, known as 5-enolpyruvylshikimate-3-phosphate synthase (EPSPS), was inhibited by the α-phellandrene, being similar to glyphosate. The molecular dynamic showed that α-phellandrene was very stable during 200 ns of evaluation. The in silico toxicological evaluation showed that all compounds are safe for the environment, but are bioaccumulative. As such, the EO of *P. acutifolium* could be used as a bioherbicide in the future. In addition, an in vivo toxicological evaluation, and an environmental study of the EO is needed to assess its safety. 

## Figures and Tables

**Figure 1 molecules-28-03348-f001:**
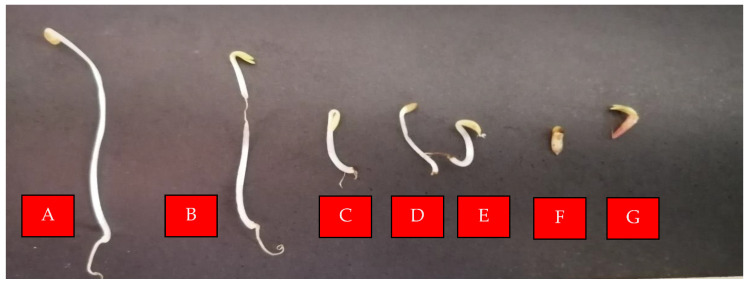
Phytotoxic activity of the essential oil of *P. acutifolium* on *Lactuca sativa* seeds. A: Negative control (DMSO 0.1%); B: *P. acutifolium* 0.1%; C: *P. acutifolium* 0.5%; D: *P. acutifolium* 1%; E: *P. acutifolium* 5%; F: *P. acutifolium* 10%; G: Glyphosate 2%.

**Figure 2 molecules-28-03348-f002:**
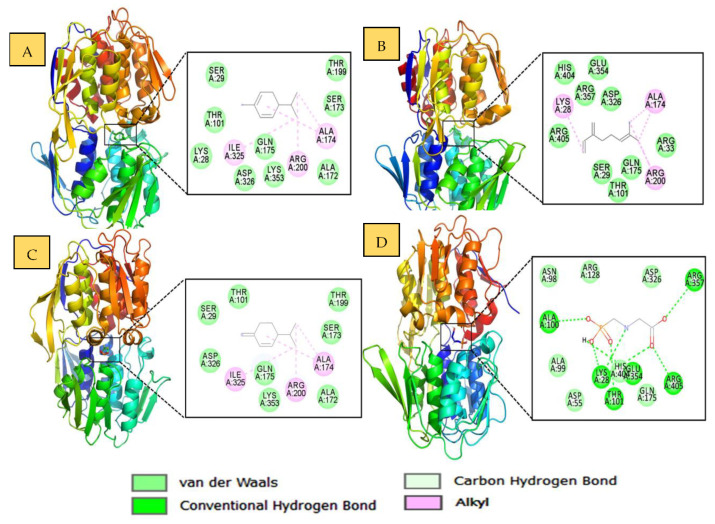
The molecular docking complex of EPSPS synthase (PDB ID: 3FJZ) with (**A**) α-phellandrene, (**B**) β-myrcene, (**C**) β-phellandrene and (**D**) glyphosate are displayed with a rendered cartoon representation on the right with a deep core accommodating the respective ligand, and on the left panel with 2D interaction plots displaying the interactions between the residues of EPSP synthase and ligands indicated by dashed lines.

**Figure 3 molecules-28-03348-f003:**
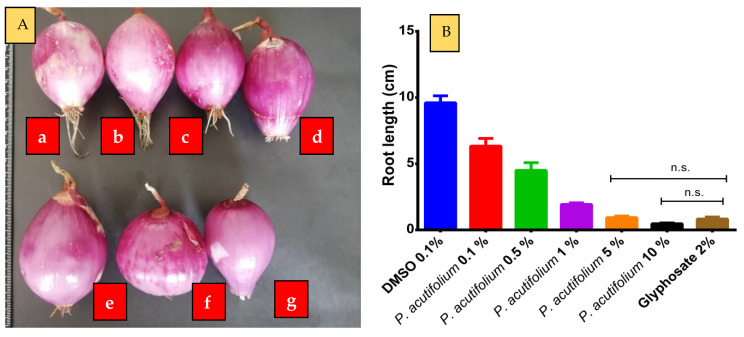
Phytotoxic activity of the essential oil of *P. acutifolium* on *Allium cepa* bulbs. (**A**): Morphology of A. cepa roots; a: Negative control (DMSO 0.1%); b: *P. acutifolium* 0.1%; c: *P. acutifolium* 0.5%; d: *P. acutifolium* 1%; e: *P. acutifolium* 5%; f: *P. acutifolium* 10%; g: Glyphosate 2%. (**B**): Root length of *A. cepa* after 10 days of treatment with EO and glyphosate. Notes: n.s.: non-significant.

**Figure 4 molecules-28-03348-f004:**
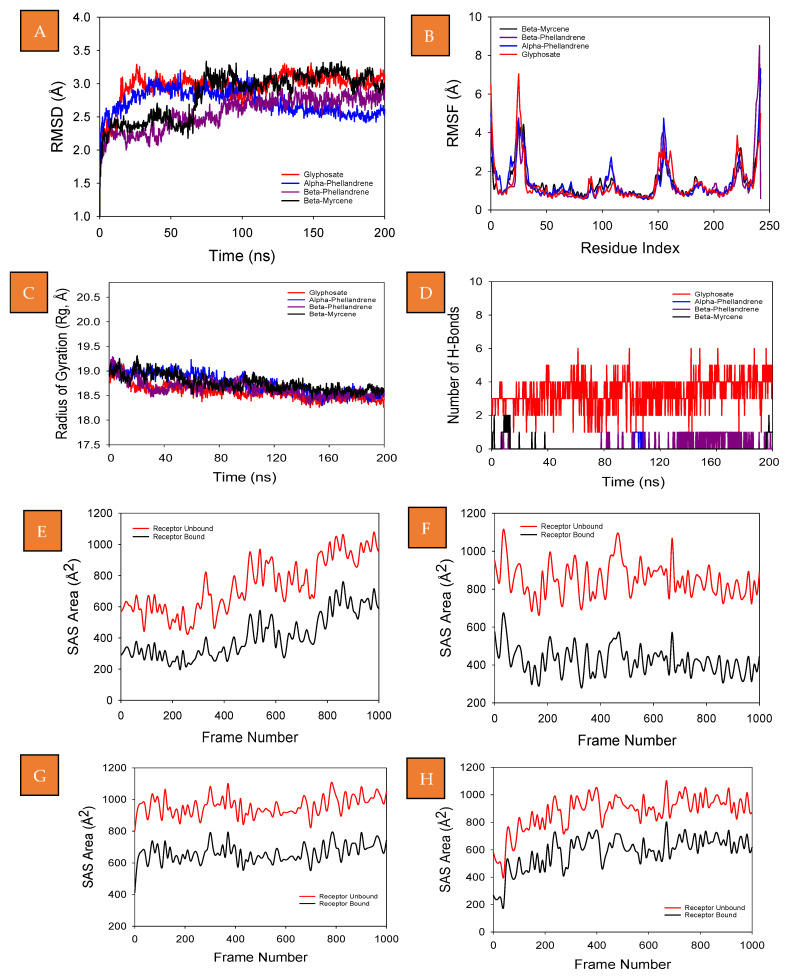
MD simulation analysis of 200 ns trajectories of (**A**) Cα backbone RMSD of 3FJZ-glyphosate (red), 3FJZ-α-phellandrene (blue), 3FJZ-β-phellandrene (purple), and 3FJZ-β-myrcene (black) (**B**) RMSF of Cα backbone of 3FJZ-glyphosate (red), 3FJZ-α-phellandrene (blue), 3FJZ-β-phellandrene (purple), and 3FJZ-β-myrcene (black) (**C**) Cα backbone radius of gyration (Rg) of 13FJZ-glyphosate (red), 3FJZ-α-phellandrene (blue), 3FJZ-β-phellandrene (purple), and 3FJZ-β-myrcene (black). (**D**) Formation of hydrogen bonds in 3FJZ-glyphosate (red), 3FJZ-α-phellandrene (blue), 3FJZ-β-phellandrene (purple), and 3FJZ-β-myrcene (black). Solvent accessible surface area plots of (**E**) 3FJZ-glyphosate, (**F**) 3FJZ-α-phellandrene, (**G**) 3FJZ-β-phellandrene, and (**H**) 3FJZ-β-myrcene.

**Table 2 molecules-28-03348-t002:** Antioxidant activity of the essential oil of *P. acutifolium*.

Antioxidant	Trolox Equivalent Antioxidant Capacity (µmol TE/g)	IC_50_ (µg/mL)
DPPH	8.10 ± 0.05	160.12 ± 0.30
ABTS	150.56 ± 8.27	138.10 ± 0.06
FRAP	64.16 ± 1.40	450.10 ± 0.05

Values are expressed as mean ± standard deviation of three determinations.

**Table 3 molecules-28-03348-t003:** Phytotoxic parameters evaluated on *Lactuca sativa* seeds. Values are expressed as mean ± SD. Groups are compared with the positive control, * (*p* < 0.001).

Groups	Seed Germination (%)	Root Length (cm)	Hypocotyl Length (cm)	Root Length/Stem Length
Negative control: DMSO 0.1%	96.33 ± 5.51 *	2.57 ± 0.06 *	6.67 ± 0.58 *	0.28
*P. acutifolium* 0.1%	36.67 ± 5.77 *	2.10 ± 0.17 *	4.67 ± 0.58 *	0.31
*P. acutifolium* 0.5%	25.33 ± 5.03 *	1.57 ± 0.06 *	3.33 ± 0.58 *	0.32
*P. acutifolium* 1%	12.33 ± 2.52 *	1.10 ± 0.10 *	1.83 ± 0.29 *	0.38
*P. acutifolium* 5%	4.00 ± 1.73	0.93 ± 0.06	1.07 ± 0.12	0.47
*P. acutifolium* 10%	0.0 ± 0.00 *	0.60 ± 0.17 *	0.63 ± 0.15 *	0.49
Positive control: glyphosate 2%	6.67 ± 2.89	1.03 ± 0.25	1.03 ± 0.25	0.08

**Table 4 molecules-28-03348-t004:** Molecular docking of the major volatile components of *P. acutifolium* and glyphosate on 5-enolpyruvylshikimate-3-phosphate synthase (EPSPS) (PDB ID: 3FJZ).

Ligand	Binding Energy (kcal/mol)	Interactions
α-Phellandrene	−5.8	Alkyl bond: Ala174, Arg200, Ile325
β-Myrcene	−4.7	Alkyl bond: Lys28, Ala174, Arg200
β-Phellandrene	−5.2	Alkyl bond: Ala174, Arg200, Ile325
Glyphosate (Control)	−6.3	Hydrogen Bond: Lys28, Thr101, Ala100, Arrg405, Arg357

**Table 5 molecules-28-03348-t005:** Binding free energy components for the 3FJZ-glyphosate, 3FJZ-α-phellandrene, 3FJZ-β-phellandrene and 3FJZ-β-myrcene calculated by MM-GBSA.

Energies (kcal/mol)	3FJZ-Beta-Phellandrene	3FJZ-Beta-Myrcene	3FJZ-Glyphosate	3FJZ-Alpha-Phellandrene
Δ*G_bind_*	−36.59 ± 2.63	−35.75 ± 2.99	−48.53 ± 4.1	−46.15 ± 1.13
Δ*G_bind_*Lipo	−13.96 ± 1.03	−11.50 ± 3.1	−19.83 ± 2.3	−13.43 ± 1.6
Δ*G_bind_*vdW	−11.10 ± 2.0	−10.63 ± 2.63	−12.68 ± 2.17	−14.160 ± 3.0
Δ*G_bind_*Coulomb	−8.12 ± 1.99	−13.66 ± 2.88	−2.14 ± 1.01	−6.22 ± 0.99
Δ*G_bind_*H_bond_	−0.41 ± 0.22	−1.87 ± 0.5	−0.06 ± 0.01	−0.62 ± 0.16
Δ*G_bind_*SolvGB	16.5 ± 1.09	60.54 ± 2.8	13.65 ± 2.27	21.2 ± 1.7
*G_bind_*Covalent	1.56 ± 1.2	4.22 ± 1.07	0.85 ± 0.5	2.66 ± 1.12

**Table 6 molecules-28-03348-t006:** In-silico analysis of toxicity of the volatile compounds of *P. acutifolium* and glyphosate. List of volatile compounds: **1.** α-pinene; **2.** β-myrcene; **3.** α-phellandrene; **4.** o-cymene; **5.** β-phellandrene; **6.** α-gurjunene; **7.** β-caryophyllene; **8.** humulene; **9.** germacrene D; **10.** bicyclogermacrene; **11.** δ-cadinene; **12.** aromadendrene.

	Toxicity			Environmental Toxicity			
#	HH	AMES Toxicity	Carcinogenicity	Bioconcentration Factors	IGC_50_	LC_50_FM	LC_50_DM
**1**	0.196	0.002	0.056	2.986	4.327	5.287	5.948
**2**	0.61	0.025	0.802	2.021	4.471	5.331	5.45
**3**	0.76	0.011	0.344	2.36	3.08	3.674	4.176
**4**	0.03	0.032	0.456	2.231	3.598	4.033	4.107
**5**	0.269	0.024	0.874	2.154	4.09	4.242	4.328
**6**	0.174	0.007	0.048	3.263	4.796	5.861	6.647
**7**	0.26	0.012	0.357	3.129	4.024	4.957	6.196
**8**	0.182	0.002	0.938	3.083	3.323	4.45	5.997
**9**	0.411	0.012	0.9	2.864	4.253	4.966	5.405
**10**	0.582	0.005	0.062	3.262	4.214	6.018	6.783
**11**	0.172	0.018	0.229	3.174	3.562	5.018	5.582
**12**	0.256	0.011	0.043	3.295	4.705	5.41	6.639
Glyphosate	0.184	0.02	0.041	0.151	2.351	3.794	3.503

Notes: HH: The human hepatotoxicity. IGC_50_: Concentration of the test chemical in water in mg/L that causes 50% growth inhibition to *Tetrahymena pyriformis* after 48 h expressed as −log10[(mg/L)/(1000*MW)]. LC_50_FM: Concentration of the test chemical in water in mg/L that causes 50% of fathead minnow to die after 96 h expressed as −log10[(mg/L)/(1000*MW)]. LC_50_DM: Concentration of the test chemical in water in mg/L that causes 50% of *Daphnia magna* to die after 48 h expressed as −log10[(mg/L)/(1000*MW)].

## Data Availability

The data is available upon reasonable request.

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
