# Peer review of "Volatile Components, Antioxidant and Phytotoxic Activity of the Essential Oil of *Piper acutifolium* Ruiz & Pav. from Peru"

_molecules, 2023, doi:10.3390/molecules28083348_

Round 1

Reviewer 1 Report

One thing forgotten to add was about Figure 4 - on figure 4B, C, E, F the font is a tad small compared to 4A and D. Recommend increasing the font size for those figures. 

Author Response

Volatile Components, Antioxidant and Phytotoxic Activity of The Essential Oil of Piper acutifolium Ruiz & Pav. from Peru

By Cuadros-Siguas, et al 2023

Brief Summary:

The aim of the paper is 1) to determine the volatile components of the essential oil (EO) of Piper acutifolium from Peru, 2) determine antioxidant activity of the EO, 3) determine phytotoxic capabilities of the EO, and 4) determine if the extracted compounds within the EO have any toxicity, both towards humans and environment.

Overall I think the authors do a relatively good job determining the items listed above; however, I think the authors need to do a better job in their comparisons with other work in the results and discussion section, mainly because this genus is so huge and the plant species listed seem random. Are these plant species compared closely related in a phylogentic study? Are these species ones that are commonly found in Peru? I imagine there are a great number of Piper species found in this area. One of the example suggestions may be for the authors to use the fact that P. acutifolium EO was largely (92.4%) monoterpene rich and yet the authors do not examine or compare with plant species that are considered to be monoterpene rich. For example, there are some plant species listed by compound type in the following reference and this may be a better way to examine and compare this very large genus of plants.

Reference: Salehi et al. Piper species: A comprehensive review on their phytochemistry, biological activities and applications. Molecules. 2019, 24(7), 1364; https://doi.org/10.3390/molecules24071364

This reference has the following species as monoterpene rich:

  • Piper demeraranum: limonene, sabinene, β-pinene and α-
  • Piper chimonanthifolium: piperitone
  • Piper cubeba: sabinene and 1,8-cineole

Based on this list of compounds, the species found in Peru has a very different composition in dominant VOCs. This is just an idea and way to do a comparison that makes a bit more sense than the many different species listed. I have another option down in that section below.

One of the biggest issues I had with the paper is the inconsistency in chemical naming. Sometimes the Greek letters are spelled out in the chemical formulas, such as in Table 1 and some of the sections; other times the Greek letters are symbols, such as in Table 4 and the legends of many of the figures. Please be consistent. I recommend based on my reading of

Molecules that the authors use the symbols and replace all spelled out words (alpha, beta, etc) with the Greek symbols (α, β, δ, etc). I also ask that the authors replace capital letters and use all lowercase letter for the chemical names. This is a problem throughout the paper – lowercase mixed with uppercase. Be consistent.

Throughout the paper there is also a plethora of its or pronouns without context within the sentence they are listed within. Please be careful when using pronouns without actually having the noun the pronouns refer to within that sentence.

R1: Thank you for your valuable comments and suggestions with our manuscript to improve and get a good presentation to readers. All changes were highlighted with yellow color and light blue color because some changes were similar to other reviewer .

Article Specifics

Abstract

  1. Line 31 – please add Spectrometry after “Gas Chromatography-Mass”.

R2: It was changed (line 31).

  1. Line 32 – because the research is specifically on three radicals, do not use “such as” or “like” within the paper. This is one such occasion – these are radicals and they were tested. State that. So this sentence should read “…the reaction with three organic radicals (DPPH, ABTS, and FRAP).”

R2: It was changed (line 31).

  1. Lines 33-34 – rephrase

Original line: “Finally, the phytotoxic study was tested under two models, Lactuca sativa seeds and Allium cepa bulbs.” to “Finally, the phytotoxic capabilities of the EO was tested on two model plants, Lactuca sativa seeds and Allium cepa bulbs.

R3: It was changed (line 33-36).

  1. Lines 34-35, correct the capitalization of the VOC and fix the Greek letter/symbol issue in the following sentence and throughout the rest of this The suggestion below also shortens the sentence and makes the formatting more consistent.

Original:

“As result, the analysis identified alpha-Phellandrene as its main volatile chemical with 38.18%, followed by beta-Myrcene (29.48%) and beta-Phellandrene (21.88%).”

Suggested:

“GC-MS analysis identified α-phellandrene as main volatile chemical (38.18%) in the EO, followed by β-myrcene (29.48%) and β-phellandrene (21.88%).”

R4: It was changed (line 35-36).

  1. Lines 38-39, fix “seeds germination” to “seed germination” and “roots length” to “root length”

Original: “…against L. sativa seeds germination, inhibition of roots length and hypocotyl length. Additionally in A. cepa bulbs, the inhibition roots length…”

Suggested: “…against L. sativa seed germination, inhibition of root and hypocotyl length. Additionally in A. cepa bulbs, inhibition of root length…”

R5: R2: It was changed (line 38-39).

  1. Line 41, redundant phrasing – an enzyme is a protein, so I don’t believe that synthase and protein are needed, but regardless, synthase is not needed twice within this sentence. Original: “… 5-enolpyruvylshikimate-3-phosphate synthase (EPSPS) synthase protein (PDB ID: 3FJZ)…”

Suggested:

“… 5-enolpyruvylshikimate-3-phosphate synthase (EPSPS; PDB ID: 3FJZ)…”

R6: It was changed (line 41-42).

  1. Line 42 – correct the capitalization in the VOC: “α-Phellandrene” to “α-”

R7: It was changed (line 42).

Key words – suggest adding “GC-MS” and “Piper acutifolium” and switching “volatile oil” to

“essential oil”. Most researchers will search for essential oil or VOCs rather than volatile oil.

R7: According to your suggestions, those key words were added.

Introduction

  1. One of the references [12] in lines 124-126, may be better moved to the introduction as it explains why hydro distillation was chosen over another I think the reference gets lost within section 2.2. But I am not sure if moving the reference would help or if the reference and information needs a better explanation where it is currently located.

R8: According to your comments, we added this reference but within of this paragraph referred to the antioxidant activity. (lines 77-90).

“Antioxidants are molecules that can react with free radicals reducing their oxidizing power, causing oxidative stress and slowing or retarding oxidation. These oxidizing agents are very harmful when there is an imbalance in the oxidation-reduction system and can affect lipids, proteins, and carbohydrates in organisms [9]. Regarding the antiox-idants and the phytotoxic activity, exist an inverse relationship because the toxicity in-duced in weeds by bioherbicides can be directly associated with the generation of reactive oxygenated species to produce damage [10]. However, in recent years, the antioxidant ac-tivity of essential oils is being studied to prevent food of oxidation during storage, food stabilizers, active packaging and edible coatings in the food industry [11]. Thereby, the in-vestigation of antioxidants in essential oils seems be focused on the inhibition of lipid pe-roxidation, free radicals, or chelating metal ions, which are process involved with several pathological conditions in humans. Furthermore, depending on its extraction method, and other pretreatments prior to the distillation process, the antioxidant activity can give different results [12].  “

  1. Lines 62-64, there is an odd phrasing Original:

“On the contrary, essential oils rich in monoterpenes have shown toxicity assessed by in vitro and in vivo studies, so their use for human use should be strictly controlled by rigorous toxicities studies. The phytotoxic effect from EOs has been related with the presence of its volatile components such as carvacrol, camphor, α-pinene….”

Suggested:

“On the contrary, essential oils rich in monoterpenes have shown toxicity, assessed in vitro and in vivo studies, so their use should be strictly controlled by rigorous toxicity studies. The phytotoxic effect of EOs has been associated with the presence of volatile compounds, such as carvacrol, camphor, α-pinene….”

R9: It was changed (lines 62-67).

  1. Lines 67-69; Phrasing issues

Original: “The herbicides can be selective (2,4-dichlorophe-noxyacetic acid) or non-selective (glyphosate) or also could work during different stages of plant development [7]. Numerous biochemical processes contribute to its herbicidal…”

Suggested: “The herbicide can be selective (2,4-dichlorophe-noxyacetic acid), be non- selective (glyphosate), or work during different stages of plant development [7]. Numerous biochemical processes contribute to herbicidal…”

R9: It was changed (lines 69-71).

  1. Line 74 – there is no need for “On the other hand” This should start at “In Peru,

Piper…”

R11: It was changed (lines 92).

  1. Lines 74-76 should have a citation at the end as it is very

Original: “…wash for sores and ulcers, as well as for vaginal infections. In gastritis and menstrual disorders, oral administration of two cups per day is prescribed [9]….”

Suggested: “…wash for sores and ulcers, as well as for vaginal infections [REF]; it is also recommended for gastritis and menstrual disorders with an oral administration of two cups per day [9].”

R12: It was changed and referenced (lines 92-95).

  1. Line 81 – do not italicize “spp.” in “Leishmania ” as it is not a Latin form.

R13: It was changed (lines 99).

  1. Line 81 – Nowadays is not needed in the following sentence “Nowadays there are …” Just

“There are no reported…”

R14: It was changed (lines 99-100).

  1. Lines 82-84 – remove “also” since there “therefore” at the start of this line has same meaning. Please fix the grammar in these sentences.

Original: “Therefore, it is also necessary to characterize the essential oil obtained by gas chromatography coupled to mass spectrometry (GC-MS), and determining the antioxidant activity.”

Suggested: “Therefore, it is necessary to characterize the essential oil obtained by gas chromatography coupled to mass spectrometry (GC-MS), and determine the antioxidant activity.”

R15: It was changed (lines 100-101).

RESULTS and discussion

  1. Line 91 – poorly Recommend: “The volatile organic compounds identified by GC- MS…”

R16: It was changed (lines 110).

  1. Line 92 – it is acceptable to use numerical numbers instead of spelling out the number of compounds identified here. “revealed 14 compounds (Table 1), 2 of which …”

R17: It was changed (lines 110).

  1. Lines 92-94 – correct the chemical compounds and use symbols as in other

R18: It was changed (lines 112-112).

Original: “The GC-MS revealed the presence of alpha-phellandrene as the major component with 38.18%, followed by beta-myrcene (29.48%) and beta-phellandrene (21.88%).”

Suggested: “The GC-MS revealed the presence of α-phellandrene as the major component (38.18%), followed by β-myrcene (29.48%) and β-phellandrene (21.88%).”

R19: It was changed (lines 111-113).

  1. Lines 94-95- the chemical compounds here are correct, but the beginning part of this sentence is unclear.

Original: “Although, there are no many reports about its composition, Lognay et al.,..”

Suggested meaning: “Although, there are not many reports about its composition, Lognay et al.,..”

R19:  It was changed (lines 113-114).

  1. Line 97 – italicize genus name Piper.

R20: It was changed (lines 116).

  1. Lines 97-111 list several different species. Again, in the overall comments above I suggest the authors may be look at sister species and what compounds are found in them or use the plant species that have been shown within Piper to be predominantly monoterpene rich and compare those. I believe the point of listing these other species is to show there are variations in the major compounds found within this genus, especially as there is only one other study on this particular plant species. Personally, I do not see a need to list every single major compound from a variety of species that are in South America.

So, another option is to simply state that there are a variety of compounds that are dominant in different Piper species and the essential oils within this genus vary from monoterpene rich EOs such as in P. aduncum with linalool (31.7%)[14] and P. augustum with α-phellandrene (14.7%)[17] to sesquiterpene rich EOs in P. demeraranum with β- elemene (33.1%) and limonene (19.3%) [12], P. arboreum with bicylogermacrene (49.5%) and P. tuberculatum with βcaryophyllene (40.2%)[14].

R21: we included your sentence according to your suggestion (lines 116-118).

“there are a variety of compounds that are dominant in different Piper species and the essential oils within this genus vary from monoterpene rich EOs such as in P. aduncum with linalool (31.7%)[14] and P. augustum with α-phellandrene (14.7%)[17] to sesquiterpene rich EOs in P. demeraranum with β- elemene (33.1%) and limonene (19.3%) [12], P. arboreum with bicylogermacrene (49.5%) and P. tuberculatum with βcaryophyllene (40.2%)[14].”

  1. Having said the above, Lines 112-115 are very I only recommend a grammar fix here in Line 112. “…Table 1 differ from other studies…”

R22: It was changed (lines 132).

  1. Table There are several things that need to happen with this table to make it easier to read.

  1. First, make sure that the title and the information within that section explains what each of the abbreviations are. Do not have them at the end. For example.

Table 1. Chemical composition of the essential oil of Piper acutifolium leaves. % is the relative percentage of each compound based on a mean of 3 determinations. LRI Ref is the linear retention index obtained from data base [19]; LRI Exp is the linear retention index calculated against n-alkanes C9–C24.”

  1. Because each GC-MS is different, please list the two retention indices first, then the RT which is often not listed because this does vary by In fact, Rt could be left out and just the two retention indices included.
  2. Add a column since the compounds within this table are referred to again in Table 6 which has the compounds listed under the table in a legend. I have suggestions for that table as well, but add a column to the left of the Compound.
  3. Headers should be as follows, in the order they
    1. # – this is the numbers that are referred to (1 through 12) used later in Table
    2. Here, there are no numbers for either Unknown I or II since they are not listed later in Table 6.
    3. Compound – there really is no need for “compound”
  • LRI Ref – many of us in the field are looking for these retention indices to help us figure out where on our column the same compound will occur.
  1. LRI Exp
  2. %
  3. Molecular Formula (molecular mass) – note because you have the mass in parentheses below the formula, please use the same type of format within that header title.
  • Chemical structure
  • Chemical group
  1. Use symbols and not the Greek letters spelled out, as in the rest of the
  2. So, below is an example of what the table would look like

#

Compound

LRI Ref

LRI Exp

%

Molecular formula (molecular

                                 mass)                                                                                                                                                

Chemical structure

Chemical group

1

α-pinene

 2       

Β-myrcene                                                                                                                                   

R23: Thank you for your suggestions, table 1 was modified.

  1. Section 2.2 – this section is organized so that results are mentioned after discussion. I recommend organizing this section so that the results of this study are presented first, then the other studies are mentioned. This may be true for other sections. Keep this in mind.

R24: Thank you for your observation a new paragraph was added and modified according to your suggestions.

“According to Table 2, the results show the antioxidant activity using three organic radicals (DPPH, ABTS and FRAP). As is presented, the best method to demonstrate antioxidant activity was in ABTS with an IC50 of 138.10 ± 0.06 µg/mL and a TEAC of 150.56 ± 8.27 µmol TE/g of EO, followed by DPPH and FRAP. Studies of antioxidant activity using the evaluated methods reveal that in non-polar plant extracts, the correlation between DPPH and ABTS is limited and decreased in comparison to polar extracts [25].”

  1. Line 121-122 rephrasing maybe to make the results clearer.

“Antioxidant activity occurred in all methods evaluated using the organic radicals DPPH, ABTS and FRAP (Table 2); however, other Piper species showed higher levels of antioxidant activity. For example, the following species showed antioxidant activity against DPPH: P. auritum (14.8…”

R25: It was changed (lines 154-162).

  1. Line 128 – unitalicize the last letter of Fruit at the beginning of this

R26: It was changed (lines 165).

  1. Lines 132-137 – Originally I was thinking these lines should start this section, but I think if the above lines (121-122) get reformatted this will work well as a concluding statement, but I do recommend some condensing and clarifying phrasing.

Original: “In this study the essential oil from P. acutifolium had low antioxidant activity and could be to the presence of monoterpene hydrocarbons in high percentage which might explain these values, some essential oils with high antioxidant activity contain oxygenated monoterpenes or sesquiterpenes in their composition [25].”

Suggested: “In this study, the essential oil from P. acutifolium had low antioxidant activity (Table 2) which could be due to the presence of monoterpene hydrocarbons in high percentage as some essential oils with high antioxidant activity contain oxygenated monoterpenes or sesquiterpenes in their composition [25].”

R27: It was changed (lines 170-174).

  1. Table 2. This is another table that because of the data mentioned within this section needs to be I suggest the following format for this table because this allows the rader to just look at the Trolox equivalent capacity or the IC50 and easily see what the results are compared to the other species listed within this section.

R28: Table 2 was modified.

Table 2. Antioxidant activity of the essential oil of P. acutifolium (mean ± standard deviation).

Antioxidant

Trolox Equivalent Antioxidant Capacity

             (µmol TE/g)                                                   

IC50

(µg/mL)

DPPH

8.10 ± 0.05

160.12 ± 0.30

ABTS

150.56 ± 8.27

138.10 ± 0.06

FRAP

64.16 ± 1.40

450.10 ± 0.05

Section 2.3

  1. Lines 139-141 are confusing. I recommend starting with line 140. See suggestion below. Mainly this is because the study does not look at insecticides or fungicides, so this sentence makes no point here. Unless the authors were thinking that more is currently known about

natural insecticides and fungicides and their use and toxicity in agriculture, but we do not know much about natural herbicides.

Original: “Currently, the use of synthetic herbicides has been considered the less toxic regarding insecticides or fungicides. However, the use of bio-herbicides based on natural products like essential oils is generating much interest in the world overall in environmental care.”

Suggested: “The use of bio-herbicides based on natural products, like essential oils, is generating much interest in the world overall in environmental care.”

R29: It was changed (lines 188-189).

  1. I have a few suggestions for this paragraph since there is an organization issue. The control is not mentioned until the end and yet lines 139-141 mention natural and synthetic herbicides, so the control needs to be brought in earlier to the discussion. So, lines 164-170 should be moved up to help the reader understand the baseline and that the products currently being used in the market have phytotoxic This is especially important later when Table 6 items are discussed – toxicity to humans and the environment. This would move these lines from the end of this paragraph and help explain better the controls within this study. In the suggested paragraph there are also some suggestions for Lines 142-145. Below is a suggested way for this paragraph to be written. Also, there is no mention of the results for A. cepa bulbs, only the comparison studies.

Original: “The positive control used in this test was glyphosate which is a synthetic herbicide that blocks the enzyme 5-enol-pyruvyl-shikimate-3-phosphate synthase (EPSPS), catalyzes the sixth step in the shikimic acid pathway reducing the aromatic aminoacids such as phenylalanine, tyrosine, and tryptophan [30], its phytotoxic activity is presented in Table 3, Fig 1 and Fig. 3 with inhibition of the seed germination, roots length, hypocotyl length and the rate root length/stem length.”

Suggested: “The use of bio-herbicides based on natural products, like essential oils, is generating much interest in the world overall in environmental care. Glyphosate is a commonly used synthetic herbicide that blocks the enzyme 5-enol-pyruvyl-shikimate-3-phosphate synthase (EPSPS) and catalyzes the sixth step in the shikimic acid pathway reducing the aromatic amino acids such as phenylalanine, tyrosine, and tryptophan [30]. It is used as a positive control in this study, and its phytotoxic activity is presented in Table 3, Fig 1, and Fig. 3 with inhibition of seed germination, root length, hypocotyl length, and the rate of root length/stem length. In this study, the evaluation of P. acutifolium EO on L. sativa seeds (Table 3) showed phytotoxicity activity at 5% and 10 % comparable to 2% glyphosate. P. acutifolium EO negatively affected the germination of L. sativa seeds (Fig. 1), depending on the concentration, while the negative control (0.1 % DMSO) did not show any phytotoxic activity. However, germination was inhibited at the highest concentration, being similar to the positive control (2% glyphosate). Root and hypocotyl length of L. sativa were inhibited when exposed to 5% and 10% EO solutions compared with the negative control (Table 3). It is known that some volatile terpenes have revealed phytotoxic activity against L. sativa seeds including α-pinene, γ-terpinene, and p-cymene from Eucalyptus grandis [26]; α-pinene and β-pinene from Pinus brutia [27]; 1,8-cineole, β-phellandrene, and α-pinene from Majorana hortensis; o-cymene, and α-pinene from Thymus vulgaris; estragole, limonene, and β-pinene from Carum carvi [28]; and β-caryophyllene, (Z)-caryophyllene, and germacrene D from Ailanthus altissima [29]. There were no significant phytotoxic effect of P. acutifolium EO on A. cepa bulbs (Fig. 2); however, studies on A. cepa bulbs indicate that terpenes have a phytotoxic effect, such as β-pinene, δ- carene, and limonene from Heterothalamus psiadioides, and β-caryophyllene, and spathulenol from Baccharis patens [29]. Abd-ElGawad et al., concluded that the phytotoxic activity of the EOs is linked to its terpenoid content, overall oxygenated terpenoid [29]. Although, some sesquiterpenes like β-caryophyllene and derivates have demonstrated phytotoxic action, in this study only aromadendrene with 0.78% was identified; this oxygenated sesquiterpene might have a synergistic phytotoxic activity on L. sativa and A. cepa bulbs. In addition, monoterpenes and sesquiterpenes extracted from EOs have shown phytotoxic effects causing anatomical and physiological changes in plants, proposed mechanisms such as accumulation of lipid globules in the cytoplasm, oxidative stress, reduction of mitochondria, and an inhibition of DNA synthesis might be involved in its phytotoxicity.”

R30: It was changed according to your suggestions (lines 188-223).

  1. Table 3 – see earlier suggestions on moving the items listed below the table to within the table’s title. Watch grammar within those sections.

R31: Table 3 was modified.

Section 2.4

  1. Throughout this section – uncapitalize the compound

R32: Thank you, compound names were uncapitalize.

  1. Line 189 – change “belong” to belonging as the group is continued into he cluster that is

R33: Thank you, compound names were uncapitalize.

  1. Figure 3 – correct the capitalized compound

R3: Thank you, it was corrected (line 241)

Section 2.4 Molecular dynamics simulation - same number as the previous section, so change this to Section 2.5.

  1. Throughout this section – uncapitalize the compound names and replace spelled out Greek letters with the Greek letter.

R35: Thank you, all compound names were corrected and uniformized.

  1. Lines 223-224 – keep formatting as previous

Suggested: “The flexibility of the residues was observed to be more in the case of α- phellandrene. For β-phellandrene and β-myrcene similar peaks were observed as that of glyphosate bound complex (Figure 4B).”

R36: Thank you, all compound names were corrected and uniformized.

  1. Line 228, Fix “…fluctuations indicate less active protein since…” to “…fluctuations indicate a less active protein, since…”

R37: Thank you, all compound names were corrected and uniformized.

  1. Lines 233 – the authors already defined radius of gyration as (Rg), so they could use this instead of putting the abbreviation in.

R38: Thank you, it was corrected. (line 291)

Original: “…of radius of gyration (Rg) from 19.0 to 18.5 Å (Figure 4C) on the other hand, lowering and stable pattern also observed…”

Suggested: “…of Rg from 19.0 to 18.5 Å (Figure 4C), while a lowering and stable pattern was also observed…”

R38: Thank you, it was corrected (line 291)

  1. Lines 236-240 – same issue

Original: “Initial lowering of gyration (Rg) indicates highly compact orientation of the protein in ligand bound state. Number of hydrogen bonds between protein and ligand suggests the significant interaction and stability of the complex. The number of hydrogen bonds between 3FJZ-Glyphosate showed significant (4) numbers and with alpha-phellandrene, beta- phellandrene and beta-myrcene single hydrogen bond was observed on an average (Figure 4D) throughout the simulation time 200 ns.”

Suggested: “Initial lowering of Rg indicates a highly compact orientation of the protein in ligand bound state. Number of hydrogen bonds between protein and ligand suggests the significant interaction and stability of the complex. The number of hydrogen bonds between 3FJZ- glyphosate showed significant (4) numbers whereas with α-phellandrene, β-phellandrene and β-myrcene, a single hydrogen bond was observed on average (Figure 4D) throughout the simulation time of 200 ns.”

R39: Thank you so much, this paragraph was corrected according to your suggestions. (line 294-297)

  1. Line 243 – this first part of this sentence is not Remove “It is clearly visible…” and start this line with “The unbound state of ligands to …”

R40: Thank you so much, this was corrected according to your suggestions. (line 298)

  1. Lines 245-247 – grammar

R41: Thank you so much, this paragraph was corrected according to your suggestions. (line 301-302)

  1. Figure 4 – fix capitalization of compounds and Greek letters as symbols for

R42: Thank you so much, it was corrected according to your suggestions.

Section entitled “Molecular mechanics generalized born surface area…” has no number. This would be 2.6 and remove capitalization of all words within the line unless they are a methodology name. Formatting check on all headers.

  1. Line 256 – has no number, if the previous section is fixed, this should now be 6.

R43: Thank you so much, the numeration was corrected.

  1. Correct capitalization of compounds and Greek letters as symbols for

44: Thank you so much, it was corrected.

  1. Table 5 - fix capitalization of compounds and Greek letters as symbols for

R45: Thank you so much, it was corrected.

Section 2. is now 2.7

  1. Line 269 – relabel now as 7.

R45: Thank you so much, it was corrected.

  1. Line 271 – capitalize

R47: Thank you so much, it was corrected. (line 314)

  1. Fix all Greek words to symbols within this

R48: Thank you so much, it was corrected.

  1. Table This is a table that needs some reformatting and organizing.

  1. This table has numbers for the compounds, and because of this I recommended these same numbers be used in Table 1.
  2. The numbers of the compounds could be moved to the title and some of notes could be moved to the title of this table. This was a recommendation earlier – please check other tables within the journal to make sure the formatting is similar.
  3. HH is listed here, but on line 270 HT is Be consistent.
  4. I recommend bolding those numbers that were significant and discussed in the text, such as the numbers of HH, carcinogenicity, and bioconcentration factors.

R49: Thank you so much, table 6 was corrected.

  1. Line 273 – after β-myrcene please put (#2) as that is the number in Table 6 that corresponds to that compound.

R50: Thank you so much, it was corrected. (line 329)

  1. Lines 275-278 – these are confusing statements as the compounds mentioned (#1, 2, 3, and 5) have lower numbers than 3.000 and not what is stated. I believe that part could be removed and just stated that these compounds fall within the bioaccumulative used by the United States (1000 L/kg<BCF<5000 L/kg).

R51: Thank you so much, it was corrected. (line 330-334)

  1. Line 283 – please correct “…might be less toxicity…” to “might have less tocicity…” or

“…might be less toxic…”

R52: Thank you so much, it was corrected. (line 337)

Materials and Methods Section

  1. Line 303 – change “…was separated of the …” to “… was separated from the…”

R53: Thank you so much, it was corrected. (line 363)

  1. Line 322 – change “…carried out by triplicate…” to “…carried out in triplicate…”

R54: Thank you so much, it was corrected. (line 384)

  1. Line 328 – change “…carried out by triplicate…” to “…carried out in triplicate…”

R55 Thank you so much, it was corrected. (line 390)

  1. Line 335 – change “…carried out by triplicate…” to “…carried out in triplicate…

R56: Thank you so much, it was corrected. (line 397)

  1. Line 338 – change “Commercially seeds….” To “Commercial seeds…”

R57: Thank you so much, it was corrected. (line 400)

  1. Line 342 – change “…as the number…” to “including number…”

R58: Thank you so much, it was corrected. (line 404)

  1. Line 348 – change “…weighting between…” to “…weighing between…”

R59: Thank you so much, it was corrected. (line 410)

  1. Line 353-354 – change “…carried out by triplicate…” to “…carried out in triplicate…”

R60: Thank you so much, it was corrected. (line 416)

  1. Lines 357-358 – recommend removing “such as” since these are the compounds being studied. They are not “… in the total composition, such as α-phellandrene… “ to “…in the total composition; these compounds were α-phellandrene…”

R61: Thank you so much, it was corrected. (line 420)

  1. Lines 375-376 – the test analysis here is not explained Recommend the following:

“To interpret the hepatotoxic, Ames’ test, and the carcinogenicity test results, they were evaluated based on probability values where values near to 1 are highly toxic [41].”

R62: Thank you so much, it was corrected. (line 420)

  1. Section 10 – correct Greek letters to symbols and remove capitalization of the compound names.

R63: Thank you so much, it was corrected.

  1. Line 386 – space at start near citation

R64: Thank you so much, it was corrected. (line 448)

  1. Section 11 – spacing issue on line 398 – really large gap between number and space.

R65: Thank you so much, it was corrected.

  1. Lines 399-408 - correct Greek letters to symbols and remove capitalization of the compound

R66: Thank you so much, it was corrected.

  1. Line 405 – correct “vander” to “van der”

R67: Thank you so much, it was corrected. (line 467)

  1. Lines 407-408 – this seems to be written like a lab manual for students to follow. Recommend For example, “To determine ΔGbind, the following equation was used: “

R68: Thank you so much, it was corrected. (Lines 469-470)

  1. Line 418 – change “…used as statistical test and a …” to “…used as statistical tests and a …”

R69: Thank you so much, it was corrected. (Line 483)

Conclusion

  1. Lines 424-427 – please rephrase due to grammar issues and emphasis lost as written. Suggest: “Despite some synthetic herbicides having harmful side effects, they are becoming

vital for effective weed management. Recent interest and demand for organic fruits, vegetables, dairy products, and beverages throughout the world, especially in industrialized nations, has led to an interest in bioherbicides and potentially a product with lower harmful side effects. As such, the EO of Piper acutifolium could be used as a bioherbicide in the future.”

R70: Thank you for your observation, it was amended according to your suggestions. (Lines 489-503).

  1. Line 429 – fix grammar and italicize Piper.

R71: Thank you so much, it was corrected.

  1. Line 430 – change “..at 5% and 10% inhibited…” to “…at, 5% and 10%, the EO oil inhibited…”

R72: Thank you so much, it was corrected.

References – please check formatting throughout as several of the references have incorrect formatting, such as1, 3, 5, 6, and many more. The dates are often repeated twice along with the journal volume and the pages.

References were modified according to molecule style.

Reviewer 2 Report

The research presented in the thesis is quite interesting, although in my opinion it needs to be clarified.

How long was the obtained oil stored at 4°C? Why were freezing conditions not used?

The methodologies for determining anti-radical properties should be clarified. Were the preparations obtained soluble in aqueous solutions of solvents?

Did the Authors take into account that the preparations obtained could contain other substances with antiradical activity? No comment on this issue.

Conclusions should be edited. They're too general.

Author Response

Dear reviewer 2

Thank you for your observations and suggestion to improve our manuscript:

  1. The research presented in the thesis is quite interesting, although in my opinion it needs to be clarified.

R1. Al changes were corrected and re-written in some paragraph which was not cleared. Thank you in advance.

  1. How long was the obtained oil stored at 4°C? Why were freezing conditions not used?

R2. Thank you for your observation. Th essential oil was kept at 4°C during one week, because the experimental procedure was immediately carried out in the lab. According to Farahbakhsh et al., best results of the main volatile compounds in EO were obtained from 4°C and -20°C. whereas there was a higher decrease in the oil quality during the storage period of three months in room temperature. Although, it could vary, depending on EO and chemical compound.

Reference: Farahbakhsh J, Najafian S, Hosseinifarahi M, Gholipour S. The effect of time and temperature on shelf life of essential oil composition of Teucrium polium L. Nat Prod Res. 2022 Jan;36(1):424-428. doi: 10.1080/14786419.2020.1771711. Epub 2020 Jun 5. PMID: 32498558.

  1. The methodologies for determining anti-radical properties should be clarified. Were the preparations obtained soluble in aqueous solutions of solvents?

R3.  Thank you for your observation. In this case we used methanol to dilute the samples and we did not have any problem in the solubility.

  1. Did the Authors take into account that the preparations obtained could contain other substances with antiradical activity? No comment on this issue.

R4: In effect the EO could contain other chemical substances non-volatile. However, the main compound determined by GC-MS was a monoterpene and we might attribute to this compound the moderate antioxidant activity. Maybe, it is necessary to use other equipment like hplc to determine those compounds which were not determined by GC-MS but it would be in traces

  1. Conclusions should be edited. They're too general.

R5: Thank you, our conclusion was improved,

Reviewer 3 Report

1. The English need improvement since there are some grammatical and syntax errors in the manuscript. For example,

·         in line number 28, the word “infusion” may be as “an infusion”;

·         in line number 31, “to a” as “into a”;

·         in line number 40, “positive” as “a positive”;

·         in line number 44,  “bioherbicide” as “a bioherbicide”;

·         in line number 63, “related with” as “related to”;

·         in line number 114, “the responsible” as “responsible”;

·         in line number 114, “of the” as “for the”;

·         in line number 121, “the table” as “table”;

·         in line number 133, “and could” as “which could”;

·         in line number 143, “comparable” as “was comparable”;

·         in line number 144, “depending of” as “depending on”;

·         in line number 147, “Root” as “The root”;

·         in line number 187 and 195, “lowest” as “the lowest”;

·         in line number 247, “acceptable” as “the acceptable”;

·         in line number 215, “Stable” as “The stable”;

·         in line number 224, “in case” as “in the case”;

·         in line number 236, “highly” as “a highly”;

·         in line number 237, “Number” as “A number”;

·         in line number 241, “pattern” as “the pattern”;

·         in line number 243,  “the figure” as “figure”;

·         in line number 244, “high” as “a high”;

·         in line number 245, “unbound” as “the unbound”;

·         in line number 255, “ligand” as “a ligand”;

·         in line number 276, “more” as “of more”;

·         in line number 279 and 283, “the glyphosate” as “glyphosate”;

·         in line number 303, “separated of” as “separated from”;

·         in line number 322, 328 and 355, “out by” as “out in”;

·         in line number 343, “and 2%” as “and a 2%”;

·         in line number 349, “were placed” as “have placed”;

·         in line number 351, “diluent” as “the diluent”;

·         in line number 356, “insight on” as “insight into”;

·         in line number 360, “standard” as “a standard”;

·         in line number 365, “form” as “to form”.

The grammar mistakes which are not mentioned here are also to be checked and corrected properly.

2. There are some typing mistakes as well, and authors are advised to carefully proof-read the text.  For example,

·         in line number 27, the word “belong” may be as “belongs”;

·         in line number 29, “study the” as “study, the”;

·         in line number 39, “Additionally” as “Additionally,”;

·         in line number 84,  “determining” as “determine”;

·         in line number 102, “leaves” as “leave”;

·         in line number 145, “not showed” as “not show”;

·         in line number 148, “exposed” as “exposure”;

·         in line number 161, “have showed” as “have shown”;

·         in line number 167, “aminoacids” as “amino acids”;

·         in line number 231 and 236, “ligand bound” as “ligand-bound”;

·         in line number 362, “kept in” as “kept at”;

·         in line number 357, “volatiles” as “volatile”;

·         in line number 377, “toxics [41] .” as “toxic [41].”;

·         in line number 403, “a one-step” as “one-step”;

·         in line number 405, “vander Waals” as “van Der Waals”;

·         in line number 408, “into” as “in”.

The typos not mentioned here are also to be checked and corrected properly.

3. Check the abbreviations throughout the manuscript and introduce the abbreviation when the full word appears the first time in the abstract and the remaining for the text and then use only the abbreviation (For example, EO, DPPH, ABTS, FRAP, IC50, radius of gyration (Rg), etc.,). Make a word abbreviated in the article that is repeated at least three times in the text, not all words  to be abbreviated.

4. The full form of the species should be given when the first time appears in both the abstract and in the remaining part of the manuscript and it should be followed by only the first letter of the genus (For example, Piper acutifolium when the first time appear and followed by P. acutifolium). Similarly, for “Streptococcus pyogenes” an other species used in the manuscript. The genus name used in the manuscript should be starts with capital letter all over the manuscript and also should be italic.

5. The introduction part appears less informative about the antioxidant imbalance and its subsequent health consequences, thus this section should be indicated as detailed to understand the manuscript in clear.

6. In the results, the authors should give the results of each concentration (0.1; 0.5; 1.0; 5.0 and 10.0%) tried in different in vitro antioxidant activity since it is not clearly highlighted in the results section .

7. The technical terms (Latin Phrase) “in vivo” (in line number 287) should be italic and it should be checked all over the manuscript. Similarly for “in-silico” (in line numbers, 140, 258, 430 etc.,)

8. The authors should include which in vitro antioxidant activity gives a better results among all the methods tried in the present investigation and also according the discussion also modified and the same should be highlighted in the conclusion section.

9. In materials and methods, plant authentication may be included with voucher number under the heading “Plant Material”.

10. The authors may include the details of the quantity of fresh sample used and the quantity of the powder obtained (the ratio between fresh plant and powder) under the heading “Plant Material”.

12. The authors should properly mention the subscript and superscript (For example, in “Na2SO4” “2” should be properly subscript and in “ABTS. +” “.+” should be properly superscript) and it should be checked throughout the manuscript, wherever applicable.

12. The statement of Lipinkski’s rule of five may be added in materials and methods, since traditionally, therapeutics have been small molecules that fall within the Lipinski's rule of five. 

13. The conclusion seems very simple. All conclusions must be convincing statements on what was found to be novel, impact based on the strong support of the data/results/discussion. Moreover, the authors may also be included the limitation of the present findings for a better understanding of the manuscript.

Author Response

Dear Reviewer 3

According to your suggestions, I am very happy with your observations to improve our manuscript:

All changes were highlighted in light blue color in your comments within the manuscript:

  1. The English need improvement since there are some grammatical and syntax errors in the manuscript. For example,
  • in line number 28, the word “infusion” may be as “an infusion”;

It was changed (line28)

  • in line number 31, “to a” as “into a”;

It was changed (line31)

  • in line number 40, “positive” as “a positive”;

It was changed (line41)

  • in line number 44, “bioherbicide” as “a bioherbicide”;

It was changed (line44)

  • in line number 63, “related with” as “related to”;

It was changed (line65) by “been associated with”

  • in line number 114, “the responsible” as “responsible”;

It was changed (line 134)

  • in line number 114, “of the” as “for the”;

It was changed (line 134)

  • in line number 121, “the table” as “table”;

It was changed (line 154)

  • in line number 133, “and could” as “which could”;

It was changed (line 171)

  • in line number 143, “comparable” as “was comparable”;

It was changed (line 196)

  • in line number 144, “depending of” as “depending on”;

It was changed (line 197)

  • in line number 147, “Root” as “The root”;

It was changed (line 199)

  • in line number 187 and 195, “lowest” as “the lowest”;

It was changed (line 241, 242)

  • in line number 247, “acceptable” as “the acceptable”;

It was changed (line 269)

  • in line number 215, “Stable” as “The stable”;

It was changed (line 264)

  • in line number 224, “in case” as “in the case”;

It was changed (line 279)

  • in line number 236, “highly” as “a highly”;

It was changed (line 290)

  • in line number 237, “Number” as “A number”;

It was changed (line 291)

  • in line number 241, “pattern” as “the pattern”;

It was changed (line 296)

  • in line number 243, “the figure” as “figure”;

It was changed 298

  • in line number 244, “high” as “a high”;

It was changed (line 298)

  • in line number 245, “unbound” as “the unbound”;

It was changed (line 299)

  • in line number 255, “ligand” as “a ligand”;

It was changed (line 300)

  • in line number 276, “more” as “of more”;

It was changed (line 331)

  • in line number 279 and 283, “the glyphosate” as “glyphosate”;

It was changed (line 334)

  • in line number 303, “separated of” as “separated from”;

It was changed (line 364)

  • in line number 322, 328 and 355, “out by” as “out in”;

It was changed (line 385, 391, 398)

  • in line number 343, “and 2%” as “and a 2%”;

It was changed (line 406)

  • in line number 349, “were placed” as “have placed”;

It was changed (line 412)

  • in line number 351, “diluent” as “the diluent”;

It was changed (line 414)

  • in line number 356, “insight on” as “insight into”;

It was changed (line 419)

  • in line number 360, “standard” as “a standard”;

It was changed (line 423)

  • in line number 365, “form” as “to form”.

It was changed (line 427)

The grammar mistakes which are not mentioned here are also to be checked and corrected properly.

All changes were amended according to your suggestions. Thank you so much

  1. There are some typing mistakes as well, and authors are advised to carefully proof-read the text. For example,
  • in line number 27, the word “belong” may be as “belongs”;

It was changed (line 27)

  • in line number 29, “study the” as “study, the”;

It was changed (line 29)

  • in line number 39, “Additionally” as “Additionally,”

It was changed (line 40)

  • in line number 84, “determining” as “determine”;

It was changed (line 102)

  • in line number 102, “leaves” as “leave”;

It was not changed, because the reference refers leaves.

  • in line number 145, “not showed” as “not show”;

It was changed (line 198)

  • in line number 148, “exposed” as “exposure”;

It was changed (line 200)

  • in line number 161, “have showed” as “have shown”;

It was changed (line 218)

  • in line number 167, “aminoacids” as “amino acids”;

It was changed (line 191)

  • in line number 231 and 236, “ligand bound” as “ligand-bound”;

It was changed (line 291, 297)

  • in line number 362, “kept in” as “kept at”;

It was changed (line 415)

  • in line number 357, “volatiles” as “volatile”;

It was changed (line 420)

  • in line number 377, “toxics [41] .” as “toxic [41].”;

It was changed (line 440)

  • in line number 403, “a one-step” as “one-step”;

It was changed (line 466)

  • in line number 405, “vander Waals” as “van Der Waals”;

It was changed (line 468)

  • in line number 408, “into” as “in”.

It was changed (line 471)

The typos not mentioned here are also to be checked and corrected properly.

  1. Check the abbreviations throughout the manuscript and introduce the abbreviation when the full word appears the first time in the abstract and the remaining for the text and then use only the abbreviation (For example, EO, DPPH, ABTS, FRAP, IC50, radius of gyration (Rg), etc.,). Make a word abbreviated in the article that is repeated at least three times in the text, not all words to be abbreviated.

R3:  All abbreviations were amended within the manuscript. Some abbreviations were erased because they are unnecessary.

  1. The full form of the species should be given when the first time appears in both the abstract and in the remaining part of the manuscript and it should be followed by only the first letter of the genus (For example, Piper acutifolium when the first time appear and followed by P. acutifolium). Similarly, for “Streptococcus pyogenes” an other species used in the manuscript. The genus name used in the manuscript should be starts with capital letter all over the manuscript and also should be italic.

R4: Thank you for your observations, those scientific names were corrected.

  1. The introduction part appears less informative about the antioxidant imbalance and its subsequent health consequences; thus this section should be indicated as detailed to understand the manuscript in clear.

R5: A paragraph related to the antioxidant activity was added in the introduction (77-89). We tried to explain the antioxidant with the herbicide activity as well as its role in chronical diseases.

  1. In the results, the authors should give the results of each concentration (0.1; 0.5; 1.0; 5.0 and 10.0%) tried in different in vitro antioxidant activity since it is not clearly highlighted in the results section.

R6: Dear reviewer, I think you refer to the phytotoxic activity because those concentrations were used in this test. We added more information about the results with the statistical analysis in lines 207-211.

  1. The technical terms (Latin Phrase) “in vivo” (in line number 287) should be italic and it should be checked all over the manuscript. Similarly, for “in-silico” (in line numbers, 140, 258, 430 etc.,)

R7. All term in Latin were changed in italic.

  1. The authors should include which in vitro antioxidant activity gives a better results among all the methods tried in the present investigation and also according the discussion also modified and the same should be highlighted in the conclusion section.

R8. The results and discussion of the antioxidant activity was improved (Lines 154-159)

  1. In materials and methods, plant authentication may be included with voucher number under the heading “Plant Material”.

R9: A voucher was added in method section (line 362).

  1. The authors may include the details of the quantity of fresh sample used and the quantity of the powder obtained (the ratio between fresh plant and powder) under the heading “Plant Material”.

R10: it was added in line 363.

  1. The authors should properly mention the subscript and superscript (For example, in “Na2SO4” “2” should be properly subscript and in “ABTS. +” “.+” should be properly superscript) and it should be checked throughout the manuscript, wherever applicable.

R11: it was corrected and reviewed in each situation. Thank you so much.

  1. The statement of Lipinkski’s rule of five may be added in materials and methods, since traditionally, therapeutics have been small molecules that fall within the Lipinski's rule of five.

R12: Thank you for your consideration. However, we do not pretend an oral use of our essential oil. On the contrary, we evaluated on the external use to be considered a bioherbicide in the future. Thus, this rule was not applied and worked with other parameters of toxicity.

  1. The conclusion seems very simple. All conclusions must be convincing statements on what was found to be novel, impact based on the strong support of the data/results/discussion. Moreover, the authors may also be included the limitation of the present findings for a better understanding of the manuscript.

R13: Thank you for your observation, the conclusion was improved and highlighting important aspects of our results.

Round 2

Reviewer 3 Report

1. There are some grammatical, alignment and typographical errors are noted in the manuscript and it should be thoroughly checked and corrected throughout the manuscript. For example,

·         in line number 70, the words “be non-selective” may be as “non-selective”;

·         in line number 86, “be focused” as “to be focused”; in table heading, “data base” as “database”;

·         in line number 190,  “that blocks the that blocks the” as “that blocks the”;

·         in line number 201, “were inhibited” as “was inhibited”;

·         in line number 215, “EOs is” as “EOs are”;

·         in line number 222, “an inhibition” as “inhibition”;

·         in line number 250, “highest” as “the highest”;

·         in line number 271, “to 3 Å” as “3 Å”;

·         in line number 271, “Stable” as “A stable”;

·         in line number 273, “higher” as “the higher”;

·         in line number 276, “the compound” as “compound”;

·         in line number 287, “Radius” as “The radius”;

·         in line number 287, “compactness” as “the compactness”;

·         in line number 288, “lowering” as “a lowering”;

·         in line number 292, “ligand-bound” as “a ligand-bound”;

·         in line number 293, “suggests” as “suggest”;

·         in line number 298, “both” as “the both”;

·         in line number 298, “state” as “states”;

·         in line number 407, “number” as “a number”;

·         in line number 412, “concentration” as “concentrations”;

·         in line number 439, “[46] .” as “[46].”;

·         in line number 496, “was seemed” as “seemed or have seemed”;

·         in line number 505, “and in” as “and”;

·         in line number 506, “EO are” as “EO is or EOs are”.

2. The suggestion is not carried out properly. Read the comment carefully and rectify it. Check the abbreviations throughout the manuscript and introduce the abbreviation when the full word appears the first time in the abstract and the remaining for the text and then use only the abbreviation. For example, in the abstract the expansion is not given for the following EO, DPPH, ABTS, FRAP, IC50 and in line number 458 the full for is given for “radius of gyration” instead of Rg. These types of corrections need to be checked all other abbreviations used in the manuscript.

Author Response

Dear reviewer 3

I am very grateful for you valuable revision according to your suggestions, we correct your observations.

  1. There are some grammatical, alignment and typographical errors are noted in the manuscript and it should be thoroughly checked and corrected throughout the manuscript. For example,
  • in line number 70, the words “be non-selective” may be as “non-selective”;

R1_ It was corrected (line 70)

  • in line number 86, “be focused” as “to be focused”; in table heading, “data base” as “database”;

R1_ It was corrected (line 86) and table 1.

  • in line number 190,  “that blocks the that blocks the” as “that blocks the”;

R1_ It was corrected (line 190)

  • in line number 201, “were inhibited” as “was inhibited”;

R1_ It was corrected (line 201)

  • in line number 215, “EOs is” as “EOs are”;

R1_ It was corrected (line 215)

  • in line number 222, “an inhibition” as “inhibition”;

R1_ It was corrected (line 222)

  • in line number 250, “highest” as “the highest”;

R1_ It was corrected (line 250)

  • in line number 271, “to 3 Å” as “3 Å”;

R1_ It was corrected (line 271)

  • in line number 271, “Stable” as “A stable”;

R1_ It was corrected (line 271)

  • in line number 273, “higher” as “the higher”;

R1_ It was corrected (line 273)

  • in line number 276, “the compound” as “compound”;

R1_ It was corrected (line 276)

  • in line number 287, “Radius” as “The radius”;

R1_ It was corrected (line 287)

  • in line number 287, “compactness” as “the compactness”;

R1_ It was corrected (line 287)

  • in line number 288, “lowering” as “a lowering”;

R1_ It was corrected (line 288)

  • in line number 292, “ligand-bound” as “a ligand-bound”;

R1_ It was corrected (line 292)

  • in line number 293, “suggests” as “suggest”;

R1_ It was corrected (line 293)

  • in line number 298, “both” as “the both”;

R1_ It was corrected (line 298)

  • in line number 298, “state” as “states”;

R1_ It was corrected (line 298)

  • in line number 407, “number” as “a number”;

R1_ It was corrected (line 404)

  • in line number 412, “concentration” as “concentrations”;

R1_ It was corrected (line 412)

  • in line number 439, “[46] .” as “[46].”;

R1_ It was corrected (line 439)

  • in line number 496, “was seemed” as “seemed or have seemed”;

R1_ It was corrected (line 496)

  • in line number 505, “and in” as “and”;

R1_ It was corrected (line 70)

  • in line number 506, “EO are” as “EO is orEOs are”.

R1_ It was corrected (line 506)

  1. The suggestion is not carried out properly. Read the comment carefully and rectify it. Check the abbreviations throughout the manuscript and introduce the abbreviation when the full word appears the first time in the abstract and the remaining for the text and then use only the abbreviation. For example, in the abstract the expansion is not given for the following EO, DPPH, ABTS, FRAP, IC50 and in line number 458 the full for is given for “radius of gyration” instead of Rg. These types of corrections need to be checked all other abbreviations used in the manuscript.

R2. Thank you so much we corrected your observations and were highlighted in yellow color.
